# Effect of Metal Surface Topography on the Interlaminar Shear and Tensile Strength of Aluminum/Polyamide 6 Polymer-Metal-Hybrids

**DOI:** 10.3390/ma12182963

**Published:** 2019-09-12

**Authors:** Erik Saborowski, Axel Dittes, Philipp Steinert, Thomas Lindner, Ingolf Scharf, Andreas Schubert, Thomas Lampke

**Affiliations:** 1Materials and Surface Engineering Group, Faculty of Mechanical Engineering, Chemnitz University of Technology, D-09107 Chemnitz, Germany; 2Micromanufacturing Technology Group, Faculty of Mechanical Engineering, Chemnitz University of Technology, D-09107 Chemnitz, Germany

**Keywords:** polymer-metal-hybrid, surface pretreatment, mechanical interlocking, roughness evaluation, interlaminar shear strength, interlaminar tensile strength

## Abstract

Mechanical interlocking has been proven to be an effective bonding mechanism for dissimilar material groups like polymers and metals. Therefore, this contribution assesses several surface pretreatments for the metallic adherent. Blasting, etching, combined blasting and etching, thermal spraying, and laser structuring processes are investigated with regard to the achievable interlaminar strength and the corresponding surface roughness parameters. The experiments are carried out on EN AW-6082/polyamide 6 polymer-metal-hybrids, utilizing a novel butt-bonded hollow cylinder specimen geometry for determining the shear and tensile strength. The experimental results indicate that the surface roughness slope has a major impact on the interlaminar strength. A laser-generated pin structure is found to provide the best mechanical performance as well as the highest surface slope of all investigated structuring methods.

## 1. Introduction

Thermoplastic polymer-metal-hybrids (PMH) offer great potential for automotive applications due to their quick processability and high strength/stiffness-to-weight ratio. In this context, a key challenge is to develop cost and time efficient techniques for creating a well-adhering interface in-between both dissimilar materials. One promising approach is adhesion by micro-scale mechanical interlocking. During joining, the polymer itself is used as the adhesive, as it infiltrates the roughness features of the metallic surface and interlocks them.

For this purpose, various joining techniques can be applied to produce mechanically interlocked PMH. In most cases, metal and polymer are placed together under pressure, while the contact area between both adherents is heated up until the polymer starts to melt. Katayama and Kawahito [1] as well as Bergmann and Stambke [2] generated the thermal energy by a laser beam, whereas Mitschang et al. [3] used induction heating. Wagner et al. [4] as well as Steinert et al. [5] utilized ultrasonic oscillations for melting the polymer. Flock [6] as well as Haberstroh and Sickert [7] applied direct heat conduction to the metallic adherent. Another approach is the direct infiltration of the metallic surface with molten thermoplastic by injection molding. Ramani [8] as well as Kleffel and Drummer [9] achieved a considerable interlaminar tensile strength when employing this method.

Besides the influence of the selected joining process, the level of achievable adhesion is directly connected to the surface characteristics of the metallic adherent. Therefore, a roughly structured and undercut surface of the metallic partner drastically increases the bonding strength compared to an untreated surface. Grit blasting is the most widely used structuring method, since it provides satisfactory adhesion with low effort and can be easily implemented in industrial applications. Consequently, it is often used as a benchmark for other structuring processes. Pan et al. [10] conducted a parameter study with different abrasive particle sizes on magnesium/carbon fiber-reinforced polymer (Mg/CFRP) laminates, concluding that larger particles create a rougher surface with a slightly increased shear strength. Etching processes offer the possibility of structuring large surfaces within a short duration, usually reaching an interlaminar strength slightly below the blasted equivalent. Mitschang et al. [11] achieved good adhesion with acidic pickling in nitric acid (HNO_3_) for an aluminum/fiber-reinforced polyamide (Al/FRPA) hybrid, whereas Nestler et al. [12] obtained the best results with alkaline pickling in a sodium hydroxide (NaOH) solution for a similar Al/FRPA hybrid. Laser structuring offers a high degree of freedom in designing the roughness features. Therefore, the highest adhesion can be obtained, although this technique is usually expensive and time consuming. Heckert and Zaeh [13] compared different laser manufactured structure sizes, wherein a kerf structure with a distance and depth of 200 µm provided the best adhesion between Al and FRPA. Steinert et al. [5] presented a self-organizing pin structure with a height of approximately 40 µm and a distance between the pins of 20 µm that reached a lap shear strength that was around 2.5 times higher than that of a blasted surface in an Al/FRPA hybrid. As an additive structuring method, thermal spraying provides an irregular, rough, and undercut surface. Utilizing a NiAl5 coating, Lindner et al. [14] reported a lap shear strength that was around 1.35 times higher than that of a blasted surface within an Al/FRPA hybrid.

As the shape of the microstructure is so important to the interface properties, proper surface characterization is mandatory for predicting the possible interlaminar strength. The most commonly used characterization method is the surface roughness measurement since it is a quick, inexpensive and widely standardized approach. Chen et al. [15] investigated the relation between various roughness parameters and the achieved shear strength of a steel/bone cement joint. Spacing (correlation length *β*) and amplitude parameters (arithmetical average roughness *R_a_*) gave no accordance, whereas the root mean square slope *RΔq* that considers the relation between amplitude and spacing gave a good accordance. However, no relation between the investigated roughness parameters and the achievable tensile strength was given.

Regarding the contributions of different authors, a significant shortcoming is the missing comparability of the obtained test results due to the different test methods that have been used. Saborowski et al. [16] reported that especially the very popular lap shear test massively underestimates the shear strength for single lap joints of metal and unreinforced thermoplastics. Therefore, Saborowski et al. [17] adapted a test method which was initially proposed by Mahnken and Schlimmer [18] for testing adhesives. Thereby, butt-bonded hollow cylinders were utilized for interlaminar strength testing. The determined shear strength values were found to be way more precise than the results of the lap shear test. Moreover, tensile strength testing can be accomplished with the same specimen geometry.

The aim of this contribution is to investigate the correlation of different roughness parameters with the interlaminar shear and tensile strength of EN AW-6082/polyamide 6 (PA6) hybrids. For this reason, several state-of-the-art surface structuring methods are applied and the surface roughness parameters *R_z_* (average maximum profile height) and tan*θ* (surface roughness slope) are evaluated. Reliable strength values are obtained by using the butt-bonded hollow cylinder specimen geometry. Preliminary lap shear tests are performed to determine strength-optimized processing parameters for grit blasting, etching, and combined blasting and etching processes for the selected PMH. Optimized laser structuring as well as thermal spraying parameters are deduced from previous investigations conducted by Steinert et al. [5] and Saborowski et al. [17], respectively. The specimens are manufactured by heat conduction hot pressing. Finally, the fracture surfaces are characterized in terms of their morphology and topography in order to investigate the relation between interlaminar strength and load direction as well as the failure mode depending on the applied structuring method.

## 2. Materials and Methods

### 2.1. Materials

The investigated metal-thermoplastic hybrid consists of Ultramid^®^ B3 PA6 (BASF, Ludwigshafen, Germany) and EN AW-6082 aluminum alloy. Table 1 shows the material properties. The parameters for the PA6 are given for a humid condition, which is achieved by conditioning the material according to ISO 1110 at 343 K and 62% relative humidity. The PA6 was conditioned before testing following this standard.

### 2.2. Testing Methods

#### 2.2.1. Lap Shear Test

The lap shear specimen illustrated in Figure 1 consists of two overlapping plates of height *h* (indices: *m* = metal, *p* = polymer), width *w* and overlapping length *l_o_*. The clamping length is given by *l_c_* and the free length is given by *l_f_*.

The geometrical parameters can be seen in Table 2. The experiments were carried out utilizing an Allround-Line 20 kN testing machine (Zwick/Roell, Ulm, Germany) with a crosshead speed of 1 mm/min. Five specimens were tested for each surface treatment.

The specimen was loaded with a tensile force perpendicular to the joining zone, causing a shear stress within the interface. The lap shear strength *τ_l,max_* was calculated from the fracture force *F_max_* divided by the overlap area *A_o_*.
(1)$$\tau_{l,\max}=\frac{F_{\max}}{A_{o}}=\frac{F_{\max}}{l_{o}w}$$

#### 2.2.2. Butt-Bonded Hollow Cylinder Test

The specimen illustrated in Figure 2 consists of two butt-bonded hollow cylinders with the outer diameter *d_o_* and the inner diameter *d_i_*.

The specimen was tested with a PTT 250 K1 hydraulic testing machine (Carl Schenck AG, Darmstadt, Germany). ER40 - 472E collets according to ISO 15488 were utilized for clamping. A steel plug was put into the polymer cylinder to support it against squeezing when the collet was tightened. The geometrical parameters of the hollow cylinder specimens are listed in Table 3. Five specimens were tested for each load case and surface treatment.

For determining the interlaminar shear strength, the joint was loaded with a torsional moment, which caused an almost pure shear stress within the interface. The specimen was twisted until it fractured, with an angular velocity of 15°/min. The shear strength τ_max_ was calculated from the maximum torque *T_max_* divided by the polar section modulus *W_p_*.
(2)$$\tau_{\max}=\frac{T_{\max}}{W_{P}}=\frac{{16T}_{\max}d_{o}}{{\pi (d}_{o}{}^{4}-d_{i}{}^{4})}$$

For identifying the tensile strength, the specimen was loaded with a tensile force, causing an almost pure normal stress within the interface. The specimen was pulled until it fractured, with a crosshead speed of 0.36 mm/min. The tensile strength *σ_max_* was calculated from the maximum tensile force *F_max_* divided by the overlapping area *A_o_.*
(3)$$\sigma_{\max}=\frac{F_{\max}}{A_{o}}=\frac{{4F}_{\max}}{\pi \left(d_{o}{}^{2}-d_{i}{}^{2}\right)}$$

### 2.3. Surface Pretreatment

#### 2.3.1. Grit Blasting

The morphology of grit blasted surfaces depends on the particle type, particle size, blasting angle, blasting distance, blasting pressure and blasting time. A corundum (Al_2_O_3_) particle type, a blasting distance of 100 mm and a treatment time of 10 s were utilized. Amada and Hirose [19] as well as Mohammedi et al. [20] found that a blasting angle of 75° provided the best adhesion for thermally sprayed ceramic coatings on a metallic substrate. Since these coatings also mainly adhere by mechanical interlocking, the same blasting angle was used here. Four different particle sizes (Wiwox F120 (90–125 µm), Wiwox F54 (250–355 µm), WFA F24 (600–850 µm) and WFA F16 (1000–1400 µm)) with three different pressures (1 bar, 2 bar and 3 bar) were investigated.

#### 2.3.2. Etching

The etching processes are based on the findings of Nestler et al. [12], who identified alkaline (NaOH) and acidic (HNO_3_) treatment to provide strong adhesion between EN AW-6082 and fiber-reinforced PA6. Alkaline etching was carried out with 2% NaOH solution at 343 K. Afterwards, the sheets were dipped into 50% HNO_3_ solution at ambient temperature for 2 min to remove reaction products from the surface. For acidic etching, the sheets were dipped for 1 min into 3% NaOH solution at 323 K in order to remove the oxide layer from the aluminum. Afterwards, the sheets were treated with 50% HNO_3_ solution. Treatment times of 1 min, 3 min, 10 min, and 20 min were investigated.

#### 2.3.3. Grit Blasting and Alkaline Etching

According to the findings of Nestler et al. [12], corundum blasting (cb) with a F24 grit size at 2 bar creates a surface roughness approximately six times higher than alkaline etching in 2% NaOH solution at 343 K for 10 min. Combined corundum blasting and etching is motivated by forming small etching structures on the much coarser blasting structures. Thereby, the additional specific surface area and fracturing of the surface should further enhance the interlaminar strength. Three different particle sizes (Wiwox F54 and WFA F24/F16, 3 bar each) and a subsequent NaOH treatment (2%, 343 K, 5 min) were investigated.

#### 2.3.4. Thermal Spraying

The thermal spraying process is based on the findings of Lindner et al. [14] and Saborowski et al. [17], who identified a NiAl5 coating suitable for creating good adhesion between aluminum and PA6. The resulting surface is characterized by a high roughness as well as the formation of undercuts. A strong adhesion of the coating onto the aluminum is achieved by corundum blasting. WFA F24 Al_2_O_3_ particles are applied with a pressure of 2 bar, an angle of 75° and a distance of 100 mm. The coating is applied by electric wire arc spraying, utilizing a VisuArc 350 spraying system (Oerlikon Metco, Pfäffikon, Switzerland). The spraying parameters are summarized in Table 4.

#### 2.3.5. Laser Structuring

The laser processing of the cylinder specimens was carried out by a nanosecond laser system (Spectra Physics^®^, Santa Clara, CA, USA) with the specifications shown in Table 5.

During laser processing, the material behavior significantly depends on the energy input into the surface. In accordance with the work of Baburaj [21], pin microstructures can be manufactured by applying a defined energy input above the material-specific threshold laser fluency. In preliminary experiments reported by Steinert et al. [5], it was found that the conditions for the generation of pin microstructures prevail in the range of a laser intensity of I ≈ 3–6 J/cm². These intensities are realized by using a defocused laser spot measuring 55 µm in diameter. The energy input leads to pin structures with an average structure height of about 40 µm and a maximum structure height of about 80 µm.

### 2.4. Specimen Production

The specimens were produced by heat conduction hot pressing. Beforehand, the PA6 was dried at 343 K. This avoided the formation of interfacial cavities by evaporating water during the hot pressing process. Moreover, the metallic surfaces were ultrasonically cleaned and degreased in ethanol. Figure 3 illustrates the hot pressing tool used for producing the hollow cylinder specimens.

Note, that the lap shear specimens were manufactured using the same tool with adapted specimen holders. According to the optimized production parameters determined by Haberstroh and Sickert [7], a constant interfacial joining pressure of 0.2 MPa was chosen. The maximum joining temperature was set to 508 K, which is just slightly above the melting temperature of the polyamide, in order to prevent excessive melting. Subsequently, the copper block was cooled down with an air cooling system until the temperature dropped below 373 K. Figure 4 illustrates the interfacial temperature over time during the joining process.

The temperature in the interface was observed by a thermocouple placed inside a drill-hole slightly below the metal surface. The complete hot pressing procedure can be summarized as follows:(1)Application of joining pressure (0.2 MPa);(2)Activation of heating cartridges (480 W in total) for heat generation;(3)Deactivation of heating cartridges when joining temperature (508 K) is reached;(4)Activation of air cooling;(5)Cooling down to 373 K;(6)Removal of joining pressure;(7)Removal of joined specimen;

The joined hollow cylinder specimens are reworked by turning on the inner and outer surface in order to ensure the necessary centricity for testing.

### 2.5. Roughness Evaluation

The average maximum profile height *R_z_* as well as the roughness slope tan*θ* are evaluated in this contribution. Assuming scale-independent material behavior and complete penetration of the molten polymer into the structured surface, the interlaminar strength should also be independent of the scale. An alteration in the load-bearing cross-sectional area for one profile element is exactly balanced out by a corresponding alteration in the total number of profile elements. Consequently, *R_z_* should not show a meaningful accordance with the interlaminar strength as it is only a measure for the scale, but not for the shape, for the profile elements.

On the other hand, the slope angle θ should have a direct relation to the interlaminar strength for two reasons: Firstly, a higher slope indicates the occurrence of more roughness features in relation to the roughness profile height. This leads to an enlarged specific surface and, therefore, to more possibilities for the polymer to interlock with the metallic surface. Secondly, the slope angle is directly related to micro-friction forces between metal and polymer. Figure 5 shows a shear force *F_s_* (e.g., induced by a shear load or polymer shrinkage) pressing the polymer against the metallic roughness feature. The resulting normal force *F_n_ = F_s_* sin*θ* and the tangential force *F_t_ = F_s_* cos*θ*. An increase in θ leads to an increase in *F_n_*. Hence, the maximum friction force *µF_n_* hindering the polymer from slipping is increased. *µ* denotes the friction coefficient between the polymer and metal. Additionally, *F_t_* which forces the polymer to slip is decreased.

The roughness measurements were carried out according to ISO 4287, using a Hommel-Etamic^®^ T8000 stylus profiler (JENOPTIK AG, Jena, Germany) with a 5 µm/90° stylus tip for the lap shear specimens and a 2 µm/60° stylus tip for the hollow cylinder specimens. *R_z_* describes the average maximum profile height of the roughness features (Figure 6) within five times the sampling length *l_r_* (5*l_r_* equals the total evaluation length *l_n_*). Accordingly,
(4)$$R_{z}=\frac{1}{5}\overset{5}{\underset{i=1}{\sum }}R_{\mathrm{zi}}.$$

Assuming a simplified symmetrical wedge shape of the roughness profile, as shown in Figure 5,
(5)$$\tan\theta =\frac{{2R}_{z}}{\mathrm{RSm}}\;\mathrm{with}\;\mathrm{RSm}=\frac{1}{m}\overset{m}{\underset{i=1}{\sum }}x_{\mathrm{si}},$$
where *RSm* denotes the average width of the roughness features x_s_ within one sampling length (Figure 6).

Usually, the actual roughness profile does not consist of symmetrical, repeating wedges, but of shapes very different in terms of the horizontal and vertical extent. Therefore, ISO 4287 proposes a minimum segment length of 0.01*l_r_* and a minimum segment height of 0.1*R_z_* for the peaks that constitute one profile element. Peaks below this threshold are treated as noise and considered a part of the preceding peak. Depending on the measured *R_z_* value as well as the chosen sampling length and threshold values the resulting *RSm* value can vary within a certain range. Another approach is presented by NASA Tech Brief 70-10722 [22], where the recorded output signal from the surface roughness tester is used to calculate
(6)$$\tan\theta =\frac{1}{l_{n}}\int_{0}^{l_{n}}\left|\frac{\mathrm{dy}}{\mathrm{dx}}\right|\mathrm{dx},$$
where *y* is the profile height signal as a function of distance *x* within the evaluation length *l_n_*. This approach yields the exact same value of tan*θ* when applied to the simplified profile shape shown in Figure 5. Applied to an actual roughness profile, a clear value is received, being independent from arbitrarily chosen threshold values like those of the 2*R_z_/RSm* approach. Consequently, Equation (6) was chosen for roughness slope evaluation within this study.

## 3. Results and Discussion

### 3.1. Lap Shear Specimens

Figure 7a illustrates the results for the corundum blasted surfaces obtained by the lap shear tests. Despite considerable differences in the pressure, particle size and achieved roughness, the results only vary in a range of 9.98–12.17 MPa. The corresponding surface parameters for the highest strength values achieved with each particle size are shown in Table 6.

Despite a drastic difference in the average maximum profile height, the lap shear strength as well as the slope values are relatively close together. Except for the F24 treated surfaces, the lap shear strength increases with an increasing particle size. However, a clear correlation between the blasting pressure and strength is not observed. A higher pressure increases *R_z_* as well as tan*θ* but also forces the embedding of corundum particles into the aluminum surface. Embedded particles may come loose when a load is applied to the interface. However, they are also considered roughness features when performing the surface roughness evaluation. Hence, the interlaminar strength in relation to tan*θ* decreases.

Figure 7b shows the lap shear strength for the etched surfaces. For the alkaline etching with NaOH, the strength (8.65–10.26 MPa), *R_z_* and tan*θ* increase with the treatment time. For the acidic etching with HNO_3_, no noticeable structuring effect could be observed, and the obtainable lap shear strength is thus quite low and almost equal for all investigated treatment times.

The results for the combined blasting and etching treatment are shown in Figure 7a. The change in strength as well as in roughness values is shown in Table 7.

The additional etching leads to a considerable decrease in the lap shear strength for F54 blasted aluminum, whereas for F24 and F16, the difference is rather small. Since the difference in the maximum profile height between F54 blasted and NaOH etched aluminum is not as pronounced as for larger particle sizes, the etching causes a leveling of smaller roughness features rather than an additional structuring effect. For the F16 particle size, an additional structuring effect is more obvious, as shown in the cross-sections in Figure 8b. The small roughness features created by the etching treatment (Figure 8d) are clearly pronounced on the much larger roughness features created by the blasting treatment (Figure 8a). However, a leveling of sharp edges, as well as a loss of undercuts, are observed, leading to a decrease in the micro-clamping area. For the F16 particle size, the additional etching led to a slight decrease in the lap shear strength, whereas a slight increase is noted for the F24 particle size. Therefore, no final statement can be made on whether additional etching is beneficial to the interlaminar strength.

It is noteworthy that a completely untreated aluminum surface was not able to create adhesion to PA6. The specimens delaminated in the climate chamber. Therefore, no results can be provided.

In Figure 9a, the lap shear strength data for the different treatment conditions is related to the determined *R_z_* value. For all of the tested treatment techniques, a clear correlation of *R_z_* and the lap shear strength is not observed and both values are rather randomly distributed. Only the surfaces that show a low *R_z_* value show a correspondingly low lap shear strength (HNO_3_ treatment 1–20 min/NaOH treatment 1 min). In Figure 9b, the lap shear strength is related to the tan*θ* value. In contrast to *R_z_*, an acceptable correlation between the lap shear strength and tan*θ* is observed. However, the measured data still shows considerable scattering. Possible influencing factors are as follows:Omission of undercuts when recording the roughness profile;Embedding of corundum particles;Scale-dependent material behavior.Poon et al. [23] reported an underestimation of vertical roughness parameters and the loss of submicron details due to the stylus tip size of the surface roughness tester. Therefore, more detail in relation to the structure’s size is lost for surfaces with smaller *R_z_*, leading to a stronger underestimation of tan*θ*.Fluctuations due to the used test method, differences in meltdown in the overlapping area and in the overlapping length due to the tolerances of the sheets, and misalignment between the sheets due to play in the specimen holders have a negative influence on the overall specimen quality.

For the hollow cylinder tests, differences in geometry can be almost excluded by turning the specimens to uniform diameters. Additionally, the surface roughness measurements will be carried out with a 2 µm stylus tip size instead of a 5 µm size for increasing the precision of the deduced roughness parameters.

### 3.2. Hollow Cylinder Specimens

Figure 10 shows the interlaminar shear and tensile strength obtained from the hollow cylinder tests. The laser-generated pin structure created the best adhesion by far, followed by the NiAl5 thermal spray coating and the F16 blasted surface. The lowest strength values were achieved with the combined F16 blasting and NaOH etching treatment.

The ratio *τ_max/_σ_max_* decreases homogeneously when increasing the overall strength. When looking at the corresponding roughness values in Table 8, it is obvious that *R_z_* does not allow a prediction of the possible interlaminar strength. In example, the pin structure provides the highest interlaminar strength by far, but shows the lowest *R_z_* value.

However, tan*θ* shows much better accordance, as illustrated by Figure 11. The interlaminar shear and tensile strength increase homogeneously with tan*θ*, showing a huge gap between the NiAl5 coating and pin structure.

It is noteworthy that the actual tan*θ* of the pin structure is obviously much higher than the measured value. Assuming an average structure height of 40 µm and an average structure width of 20 µm, as shown in Figure 12d, the resulting tan*θ* according to Equation (5) would be approximately 4. Since the measuring line does not hit every peak and Equation (6) is applied to the resulting profile, the actual tan*θ* should be a bit lower, but still much higher than the measured 0.778. When comparing the measured roughness profile shown in Figure 13b with the cross-section in Figure 12d, the loss of detail due to the missing penetration of the stylus tip can clearly be seen. However, when comparing the measured profile of the F16 blasted structure in Figure 13a with the corresponding cross-section in Figure 12a, the loss of detail is far less pronounced.

When comparing the morphology of the corundum blasted surfaces in Figure 12a,b and Figure 14a,b, respectively, the indentations created by the additional etching treatment are clearly pronounced. On the other hand, there is an obvious loss of undercuts and all sharp submicron features. In particular, the loss of undercuts may cause the drastic decrease of 51% in σ_max_, whereas τ_max_ only decreased by 19%. The NiAl5 coating illustrated in Figure 12c and Figure 14c, respectively, shows a high amount of small splats that function as undercut features. Hence, the achieved interlaminar strength is higher than that obtained from corundum blasting treatment only. The laser treatment illustrated in Figure 12d and Figure 14d, respectively, causes the formation of steep, pin-like structural elements arranged in a high spatial density. Undercuts are provided by molten together or crooked pins as well as submicron roughness features of the almost perpendicular pins. In consequence, a high interlaminar strength for both the tensile and the shear load is achieved.

The fractured surfaces of the tensile as well as the shear-loaded hollow cylinder specimens are depicted in Figure 15. Dark areas indicate the residing polymer, since the images were taken using the back-scattering detector (BSD). For all the tested joints and surface pretreatment conditions, polymer residues are present at the metallic adherent. However, almost no polymer residues are present at the fractured surfaces of the blasted and subsequently etched specimens (a, e). Additionally, a higher number of coarse residues are found at the shear-loaded, just-blasted specimen (b), whereas nearly no residues are observed for the corresponding tensile load case (f). In contrast, a high number of comparably small residues are located at the fractured surface of the shear-loaded thermal spray coating (c) and likewise, significantly less residues of a once again reduced size are found for the tensile load case (g).

Generally, an increasing content of polymer residues is observed along with an increasing joint strength and fragmentation of the treated surfaces, respectively (Figure 15a–c,e–g). Further, the difference in the number of residues in-between the shear and tensile load case is explained by the orientation of the roughness features that predominantly provide undercuts against shear rather than tensile loads.

For the tested laser pin structure, the BSD images for both load cases show that an outstandingly high amount of residing polymer is located in-between the structure. Figure 16 provides a detailed view of these fractured surfaces. For the shear (a, b) and the tensile (c, d) load case, failure takes places in the polymeric adherent. However, different types of failure, like failure above the pins oriented parallel to the surface (a) as well as clearly plastically deformed polymeric residues (b–d) can be found.

## 4. Conclusions

Based on the experimental results and analyzes performed in the present work, the following conclusions can be drawn:Corundum blasting creates considerable adhesion between PA6 and EN AW-6082, even with a smaller grit size and low pressure. There is no evidence that a higher blasting pressure leads to better adhesion.A combination of corundum blasting and alkaline etching treatment decreases the adhesion compared to corundum blasting only due to the loss of undercuts and submicron roughness features.A thermally-sprayed NiAl5 coating creates better adhesion than corundum blasting.A self-organized, laser-generated pin structure creates the highest adhesion by far due to its high spatial density of roughness features.The torsion and tension test using butt-bonded hollow cylinders allows for a more meaningful determination of interlaminar strength values than the lap shear test. Additionally, all load directions can be tested with one specimen geometry.The interlaminar strength in the shear as well as tensile direction is strongly related to the surface roughness slope tan*θ*.Roughness evaluation with a stylus profiler leads to an underestimation of tan*θ* due to the missing penetration of tight roughness profile valleys as well as the loss of submicron details. This effect increases with an increasing structure density and decreasing structure size. A stylus tip size as small as possible should be used.There is no meaningful relation between the average maximum profile height *R_z_* and the interlaminar strength for structure sizes within the investigated range (1.26 µm < *R_z_* < 142 µm).The fracture analysis of the hollow cylinder specimens reveals that the interlaminar strength is strongly related to the number of polymer residues in the surface structure. A higher interlaminar strength leads to more residues. Shear testing leads to more residues than tensile testing.

## Figures and Tables

**Figure 1 materials-12-02963-f001:**
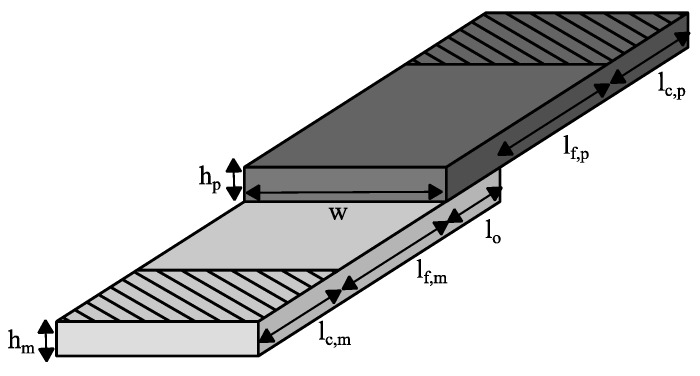
Lap shear specimen geometry.

**Figure 2 materials-12-02963-f002:**
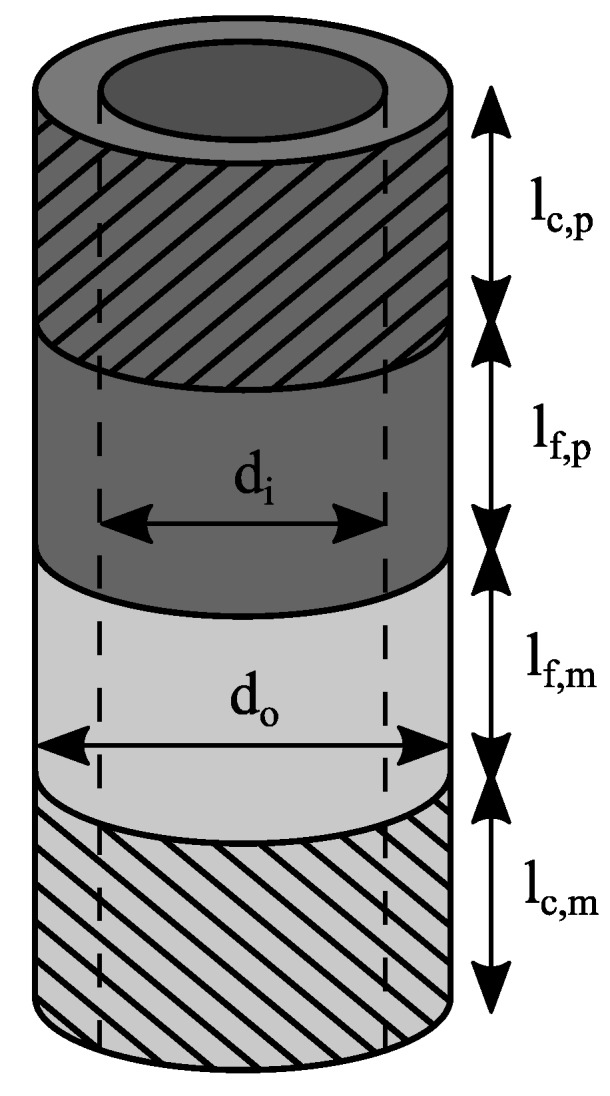
Hollow cylinder specimen geometry.

**Figure 3 materials-12-02963-f003:**
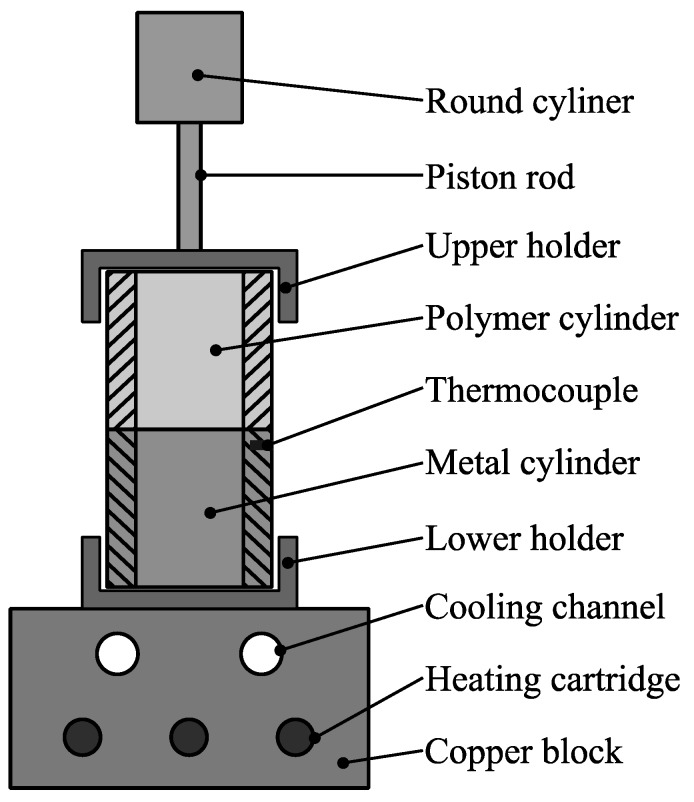
Hot pressing tool.

**Figure 4 materials-12-02963-f004:**
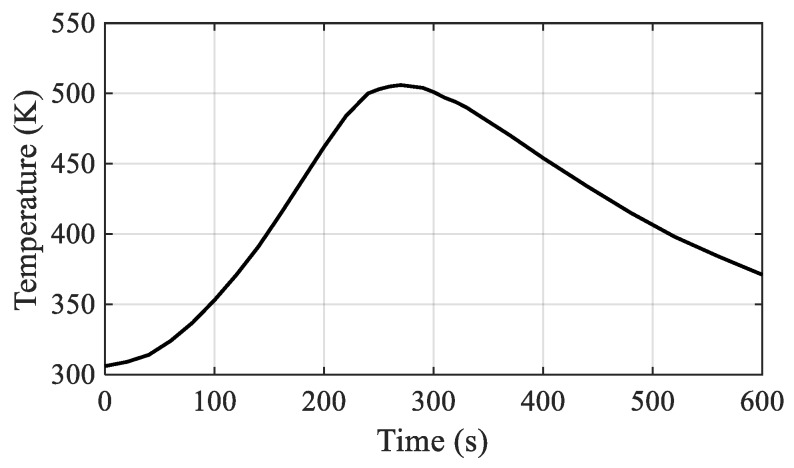
Interfacial temperature during the joining process (hollow cylinder specimens).

**Figure 5 materials-12-02963-f005:**
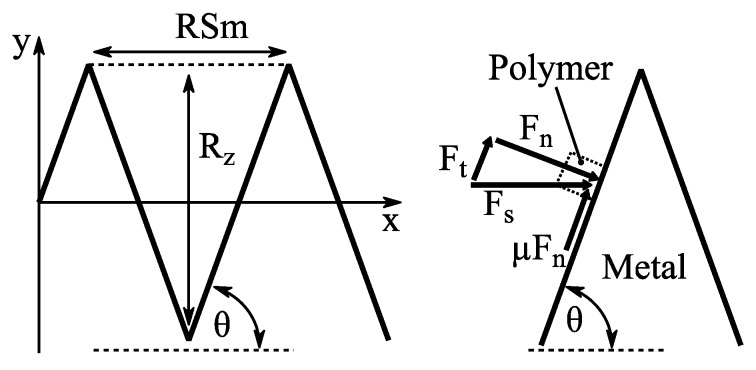
Roughness slope and resulting forces.

**Figure 6 materials-12-02963-f006:**
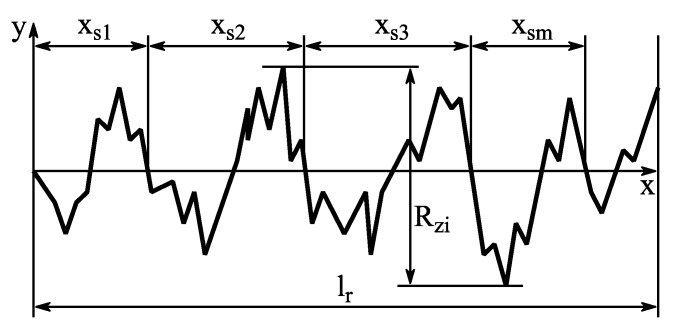
Schematic representation of the surface profile.

**Figure 7 materials-12-02963-f007:**
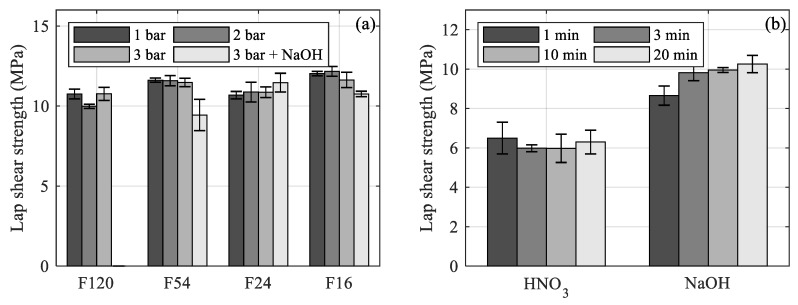
Lap shear test results for (**a**) blasting and combined treatment and (**b**) etching treatment.

**Figure 8 materials-12-02963-f008:**
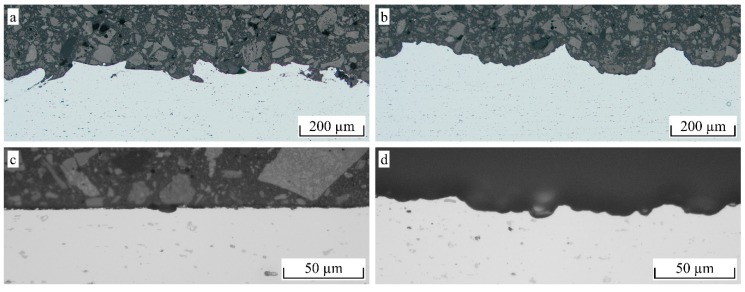
Cross-sections for (**a**) blasting (F16, 3 bar) (**b**) combined blasting (F16, 3 bar) and etching (5 min NaOH) (**c**) acidic etching (20 min HNO_3_) and (**d**) alkaline etching (20 min, NaOH) treatment applied to EN AW-6082 sheets.

**Figure 9 materials-12-02963-f009:**
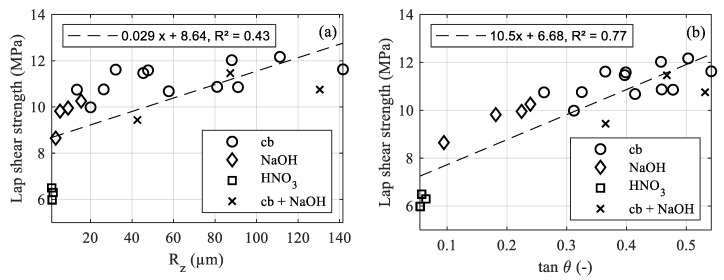
Lap shear strength (**a**) over *R_z_* and (**b**) over tan*θ*, both with a 5 µm tip diameter.

**Figure 10 materials-12-02963-f010:**
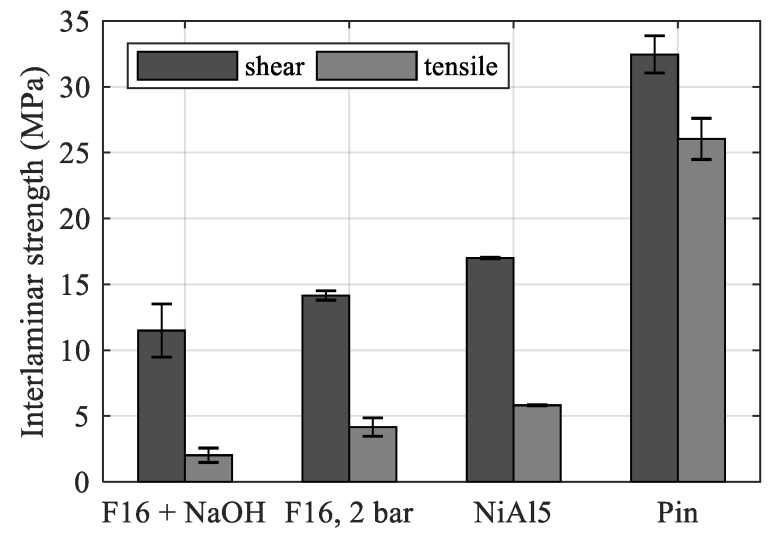
Interlaminar strength of hollow cylinder specimens.

**Figure 11 materials-12-02963-f011:**
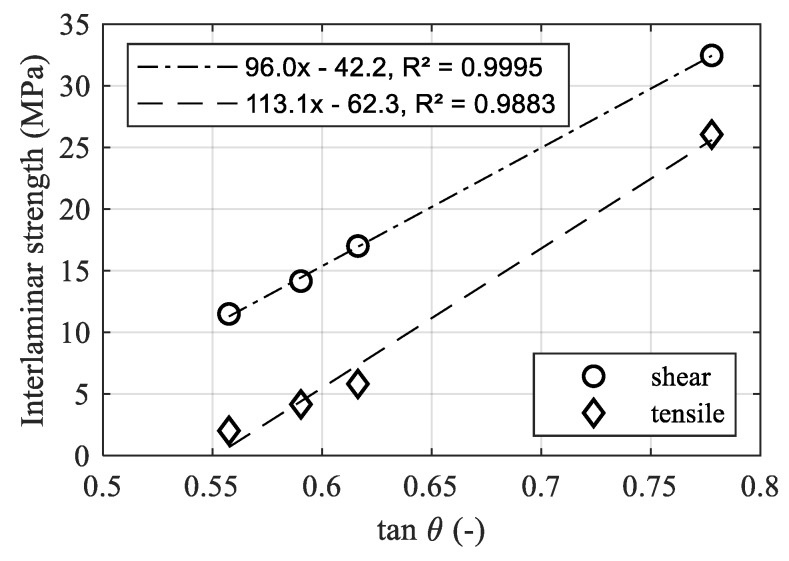
Shear and tensile strength over tan*θ* (2 µm tip diameter).

**Figure 12 materials-12-02963-f012:**
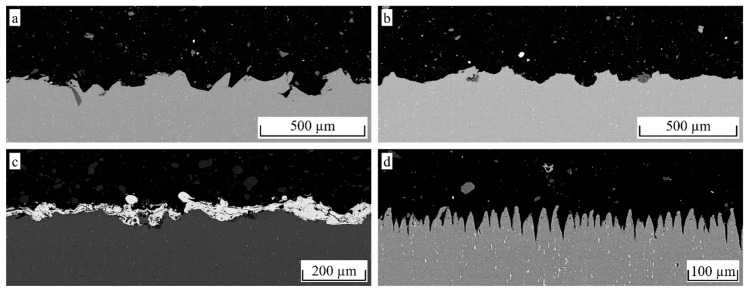
Representative cross-sectional images (SEM) of hollow cylinder specimens, scale normalized to image width/*R_z_* = 1/15 (**a**) F16, 2 bar (**b**) F16, 3 bar + 5 min NaOH (**c**) NiAl5 coating and (**d**) pin structure.

**Figure 13 materials-12-02963-f013:**
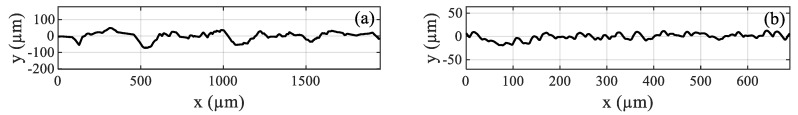
Roughness profiles (2 µm tip), equal axis scaling (**a**) F16, 2 bar and (**b**) pin structure.

**Figure 14 materials-12-02963-f014:**
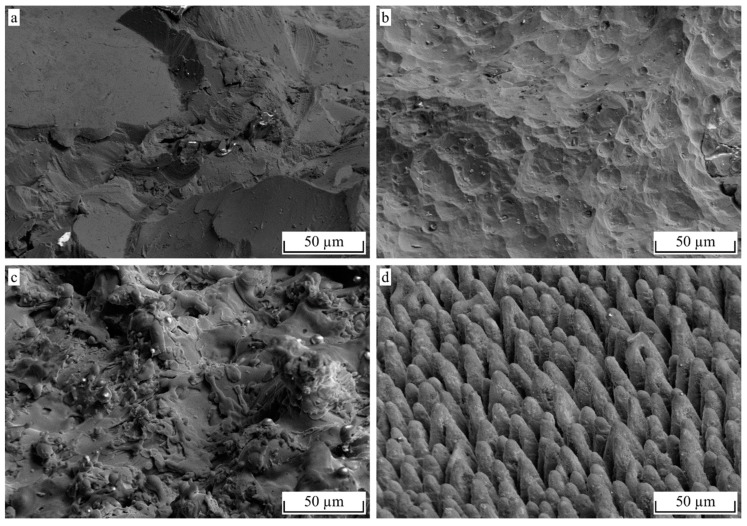
SEM images of hollow cylinder specimens, 60° tilt, 25 kV (**a**) F16, 2 bar (**b**) F16, 3 bar + 5 min NaOH (**c**) NiAl5 coating and (**d**) pin structure.

**Figure 15 materials-12-02963-f015:**
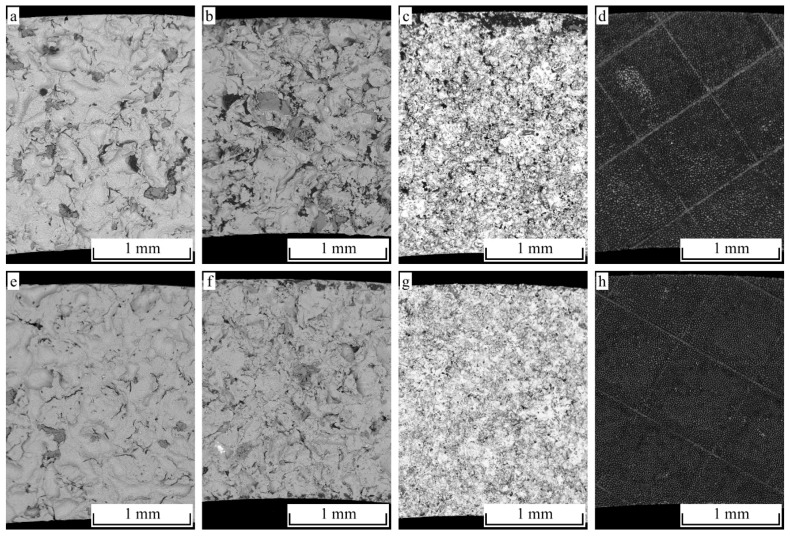
Fracture analysis (SEM-images back-scattering detector (BSD) contrast) of the tested hollow cylinder specimens. Shear- (**a**–**d**) and tensile- (**e**–**h**) loaded specimens are depicted for the differently structured surfaces: (**a**,**e**) F16, 3 bar + 5 min NaOH, (**b**,**f**) F16, 2 bar (**c**,**g**) NiAl5 coating and (**d**,**h**) pin structure.

**Figure 16 materials-12-02963-f016:**
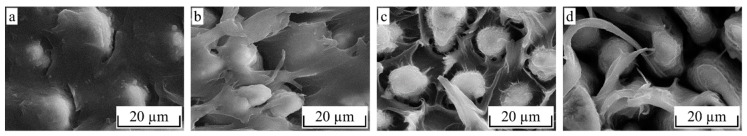
SEM images of the fractured surface of the pin-structured hollow cylinder specimens (**a**,**b**) shear load and (**c**,**d**) tensile load.

**Table 1 materials-12-02963-t001:** Material properties.

	EN AW-6082	PA6 (humid)
Density (kg/m³)	2.7	1.14
Elastic modulus (MPa)	70000	1800
Poisson’s ratio (-)	0.34	-
Yield strength (MPa)	260	60
Ultimate strength (MPa)	310	-
Elongation to failure (%)	7	200
Melting temperature (K)	933	496
Thermal expansion coefficient (10^−6^/K)	23.4	70
Thermal conductance (W/(m·K))	170–220	0.23
Specific heat (J/(kg·K))	898	1700

**Table 2 materials-12-02963-t002:** Geometrical parameters of the lap shear specimens.

w (mm)	l_o_ (mm)	l_c,m_ (mm)	l_f,m_ (mm)	l_f,p_ (mm)	l_c,p_ (mm)	h_m_ (mm)	h_p_ (mm)
25	5	45	50	40	35	3	3

**Table 3 materials-12-02963-t003:** Geometrical parameters of butt-bonded hollow cylinder specimens.

d_i_ (mm)	d_o_ (mm)	l_c,m_ (mm)	l_f,m_ (mm)	l_c,p_ (mm)	l_f,p_ (mm)
23	28	20	20	30	30

**Table 4 materials-12-02963-t004:** Feedstock material and parameters used for the thermal spraying process.

Chemical Composition	Current (A)	Voltage (V)	Spraying Distance (mm)	Air Pressure (bar)	Feed Speed (m/s)	Row Spacing (mm)
Ni–95%/Al–5%	150	30	130	3.5	0.6	5

**Table 5 materials-12-02963-t005:** Parameters used for the laser structuring process.

Laser Medium	Wavelength (nm)	Pulse Duration (ns)	Max Mean Power (W)	Focus Diameter (µm)
Nd:YVO4	532	10	13	15

**Table 6 materials-12-02963-t006:** Highest test results and roughness parameters (5 µm tip) for blasting treatment.

Treatment	τ_l,max_ (MPa)	R_z_ (µm)	tanθ
F120, 3 bar	10.76	26	0.325
F54, 1 bar	11.62	32	0.364
F24, 2 bar	10.87	81	0.459
F16, 2 bar	12.17	111	0.503

**Table 7 materials-12-02963-t007:** Change in test results and roughness parameters (blasting (3 bar) → combined blasting (3 bar) and etching (5 min, NaOH) treatment, 5 µm tip).

Treatment	τ_l,max_ (MPa)	R_z_ (µm)	tanθ
F54	11.47 → 9.44	45 → 43	0.397 → 0.365
F24	10.87 → 11.46	91 → 87	0.478 → 0.467
F16	11.63 → 10.75	142 → 131	0.542 → 0.531

**Table 8 materials-12-02963-t008:** Interlaminar strength and roughness parameters (2 µm tip) of hollow cylinder specimens.

Treatment	τ_max_ (MPa)	σ_max_ (MPa)	τ_max_/σ_max_	R_z_ (µm)	tanθ
F16 + NaOH	11.49	2.02	5.69	124	0.557
F16, 2 bar	14.15	4.16	3.40	131	0.590
NiAl5	17.00	5.81	2.93	80	0.616
Pin	32.46	26.04	1.25	46	0.778

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
