# Peer review of "Effect of Metal Surface Topography on the Interlaminar Shear and Tensile Strength of Aluminum/Polyamide 6 Polymer-Metal-Hybrids"

_materials, 2019, doi:10.3390/ma12182963_

Round 1

Reviewer 1 Report

In this article, the authors investigate the correlation of different roughness parameters 83 with the interlaminar shear and tensile strength of EN-AW6082/polyamide 6 (PA6) hybrids. Different testing methods and surface pretreatment are showed. The results and conclusions are clearly presented. In this form the article can be published.

Author Response

The authors would like to thank you for the review. We are happy that you like the paper and accepted it for publishing.

Reviewer 2 Report

Comments:

Mechanical interlocking between polymers and metals is very important for practical application. In this manuscript, the authors have examined the relationship between the interlaminar strength and several surface parameters by using aluminum alloy/ PA6 hybrid. The comparison of the results between the lap shear and butt-bonded hollow cylinder tests, the authors show that the later test gives a clear difference of surface treatment. Although we could not find the reason of the test’s sensitivity is different in this manuscript, it was interesting that the performance of interlocking was strongly correlated to the simple surface roughness parameters which may be related to the side wall area faction of vertical bump. If possible, the authors should discuss the reason related to material properties such as thermal expansion and surface crystallinity of PA. We think the crystallinity and its structure distribution near the surface region of PA is important factor to determine the order of the roughness and mechanical properties. In figure 8, what is the gray particles in PA phase? Because these sizes are seemed to be larger than the surface roughness, these particle size affect on the adhesion strength. Please add comments.

Minor comments;

In table 1, thermal expansion coefficient has unit 1/K Variable symbols should be italic. Ex. l in figures 1, 2, 5, 6 and tables 2, 3. In line 291, Fn and Ft may be wrong. Fn = Fs sinθ, Ft = Fz cosθ? In figure 3, thermocouples should be indicated because of temperature control is important for set experimental conditions. Caption in figure 14, electron scanning microscope may correspond to SEM?

Author Response

The authors would like to thank the reviewer for his valuable comments. Below, we respond to the claims.

If possible, the authors should discuss the reason related to material properties such as thermal expansion and surface crystallinity of PA. We think the crystallinity and its structure distribution near the surface region of PA is important factor to determine the order of the roughness and mechanical properties.

Polymer-metal hybrids (PMH) are complex systems due to the multiple interacting influencing factors. We completely agree, that the mechanical performance of the PMH joints is influenced by the strongly unequal coefficients of thermal expansion of polymer and metal. Indeed, for the presented criterion, we assume that a minimum bond strength is necessary to overcome separation of polymer and metal caused by the residual stresses. In the presented study, we have not investigated the extent and distribution of the thermally induced residual stresses and its influence on the bond strength. The crystallinity of the polyamide in the interfacial zone can affect the mechanical joint performance. For the used unreinforced polymer, the crystallization is strongly influenced by the cooling rate. Thus, we paid special attention to ensure using the same cool rate for the all the tested samples. Further, the formation of crystallization nuclei and growth of crystalline areas can depend on nanoscale features located at the metal surface. Additionally, the extent of surface roughness features on the micro scale and particularly on the meso scale may influence the distribution and alignment of crystalline zones, which can likewise decrease or increase the overall mechanical strength. This was not part of our investigations, but offers a very interesting potential for a future study.

In figure 8, what is the gray particles in PA phase? Because these sizes are seemed to be larger than the surface roughness, these particle size affect on the adhesion strength. Please add comments.

These particles are from the embedding to make it conductive for electron microscopy. Joined specimens are not shown here. The used polyamide has no particles in it.

In table 1, thermal expansion coefficient has unit 1/K

The correction has been made.

Variable symbols should be italic. Ex. l in figures 1, 2, 5, 6 and tables 2, 3.

Unfortunately, “Materials” does not give information in their guidelines how variable symbols should be written. They adjust the layout later in the publication process to their needs. Therefore, no changes have been made.

In line 291, Fn and Ft may be wrong. Fn = Fs sinθ, Ft = Fz cosθ?

The correction has been made.

In figure 3, thermocouples should be indicated because of temperature control is important for set experimental conditions.

The thermocouple position has been added to the figure.

Caption in figure 14, electron scanning microscope may correspond to SEM?

The term has been changed from “electron scanning microscope” to “SEM”

Reviewer 3 Report

This is a nicely written paper that systematically addresses the polymer-metal interlaminar shear/tensile strength.  The data reported will be practically useful for engineers working in this field.  I recommend the manuscript be published, subjected to the following minder revision:

Why is 75°blast angle was chosen? How many samples were tested for each case? Provide torque-angle of twist curves and stress-strain curves.

Author Response

The authors would like to thank the reviewer for his valuable comments. Below, we respond to the claims.

Why 75° blast angle was chosen?

Two references to that have been added.

How many samples were tested for each case?

5 specimens were tested for each specimen geometry and load case. A comment to that has been added.

Provide torque-angle of twist curves and stress-strain curves.

The crosshead displacement and angle differs very much from the actual material displacement and angle at the beginning of the clamping because of movement inside of the collet. Unfortunately, we did not record grey scale correlation data for the torsion and tensile test within this publication. Therefore, no correct data can be provided here. In our previous publication (Ref. [16]), force-strain and torque-shear angle curves deduced from grey scale correlation data are provided. But you can only see the material behavior of the polyamide there since it is so much softer then the aluminum.

Reviewer 4 Report

Very comprehensive and coherent article. I have only minor remarks, from which the most important one is the lack of a reference sample with non-treated surface. I strongly recommend adding results from a reference sample to the article.

What are the particle sizes of Wiwox F120/F54 and WFA F24/F16 mentioned in the section 2.3.1.? Please change the “3” from HNO3 (line 147) into sub-script instead normal size. Why there is no reference sample showing the adhesion between non-treated aluminium and the polyamide? This would give a better picture of the effect of the surface treatments. In line 267 please change “Figure 7b” into “Figure 7a” Please make clear which result belongs to which treatment in table 7.

Author Response

The authors would like to thank the reviewer for his valuable comments. Below, we respond to the claims.

What are the particle sizes of Wiwox F120/F54 and WFA F24/F16 mentioned in the section 2.3.1.?

The particle sizes have been added.

Please change the “3” from HNO3 (line 147) into sub-script instead normal size.

The correction has been made.

Why there is no reference sample showing the adhesion between non-treated aluminium and the polyamide? This would give a better picture of the effect of the surface treatments.

A completely untreated aluminum surface was not able to create adhesion to PA 6. The specimens delaminated in the climate chamber. Therefore, no results can be provided. A comment to that has been added.

In line 267 please change “Figure 7b” into “Figure 7a”.

The correction has been made.

Please make clear which result belongs to which treatment in table 7.

The table caption has been enhanced.